# OpenReview forum: "When LLM Meets DRL: Advancing Jailbreaking Efficiency via DRL-guided Search"
_NeurIPS.cc/2024/Conference — NeurIPS 2024 poster_

### Official Review · Reviewer_5U1Z · 2024-06-25

**Soundness:** 2
**Presentation:** 3
**Contribution:** 2
**Rating:** 6
**Confidence:** 4

**Summary:**

This paper proposed a black-box jailbreaking framework, RLbreaker, it use Reinforcement learning to help the optimization of jailbreaking prompt.

At training each step, the designed agent of RLbreaker will select a mutator from a small set, and then the helper LLM will use the selected mutator to enhance the prompt.  The optimized prompt will be feed to victim LLM and an un-aligned LLM, the framework will use the difference between 2 responses to calculate the reward for next training step.

The experiment section include general attack and transfer attack, as well as attack against Jailbreaking Defenses.

**Strengths:**

Compare with the most popular baselines (i.e. the GCG and AutoDAN), the framework proposed by this paper decrease the search space so that the training speed is fast.

The structure of the paper is very clear and it is easy to understand the framework.

The section 4, consists of 3 kinds of evaluation (general/transfer/ablation), is reasonable.

Although the action base is small, but in the ablation section, the results show that randomly select action will lead a significant performance drop. This proves that the agent is definitely effective.

**Weaknesses:**

All experiments are done on a 300-samples test set, and  it is the same distribution as the training set. Therefore hard to prove its robustness and effectiveness.

**Questions:**

How did you determine the action space you ultimately chose?

**Limitations:**

Yes

---

> ### Author Rebuttal · Authors · 2024-08-07
>
> **The reviewer points out that all experiments are conducted on a 300-sample test set, which has the same distribution as the training set, making it difficult to prove the method's robustness and effectiveness.**
>
> We thank the reviewer for the question. When splitting the training and testing set, we avoided putting similar templates in both training and testing to encourage enough differences and avoid data leakage. To further demonstrate the generalizability of our method, we conducted an additional experiment. We first trained our agents on the training set, following the same setup in Section 4. Then we tested our agents on the other two harmful question sets. One is from our baseline: GPTFUZZER, where they construct 100 questions from scratch. The other is called MaliciousInstruct from [2], where they claim it is a different dataset from AdvBench that also contains 100 harmful questions. We select Llama2-7b-chat as the target model and GPT-3.5-Turbo as the helper model, and the results are in Table 9 in the submitted pdf file.
>
> We can see that after applying our RL agent on the other two testing sets, the performance even gets improved, compared to the original AdvBench testing set. Furthermore, we showed the transferability of our agents across different models, which also demonstrates the generalizability and robustness of our approach.
>
>
> **The reviewer asks how the authors determined the chosen action space.**
>
> We thank the reviewer for the question. When designing the action space $\mathcal{A}$ of an RL agent for our jailbreaking attack, the requirements of the action space $\mathcal{A}$ are two. First, it should enable diverse and substantial changes to the prompts. Second,  $\mathcal{A}$ should not be ultra-large, which will make it difficult for the agent to learn an effective policy.
>
> Before designing our action space by selecting different mutators, we made an initial trial of designing the action as selecting different tokens from the whole vocabulary, and the state is the current prompt. We refer to this design as “token-level RL” and the details are in Section D.4 in Appendix. Our experiment results demonstrated that this token-level RL design cannot generate any effective jailbreaking prompts. The reason is that the action space is too huge for the agent, which is the same as the vocabulary size (about 30,000). Thus, considering the maximum length of the generated jailbreaking suffix as N, the possible combination is 30,000^N, which is unrealistic for an agent to learn some effective policy.
>
> Then, we took a different avenue and considered mutating the input prompts at the sentence level. We selected some common ways of mutating a sentence as our actions that enable enough changes to the input while constraining the total action space. Note that our framework is flexible and can incorporate more actions in the future.
>
>
> [1] Gptfuzzer: Red teaming large language models with auto-generated jailbreak prompts. Yu et al., arXiv 2024.
>
> [2] Catastrophic Jailbreak of Open-Source LLMs via Exploiting Generation. Huang et al., ICLR 2024.

---

> > ### Comment · Reviewer_5U1Z · 2024-08-11
> >
> > Thanks for your rebuttal. I will keep the Rating since it is already the highest.

---

> > > ### Author Response · Authors · 2024-08-12
> > >
> > > We thank the reviewer for supporting the paper and maintaining the highest rating. We will update our paper based on the suggestions.

---

### Official Review · Reviewer_5ZZd · 2024-07-13

**Soundness:** 3
**Presentation:** 3
**Contribution:** 3
**Rating:** 6
**Confidence:** 3

**Summary:**

This work proposes a new method to jailbreak LLM to elicit harmful respones. It adapts deep reinforcement learning to learn a policy of sampling jailbreaking operations (modifying prompt) from a predefined pool. A LLM is then used to rewrite the query prompt complying with the sampled operation. To guide the policy learning, it proposes a new reward function that compares the difference between the target model's response and a reference response. The empirical results confirm the efficacy and the efficiency of the proposed method.

**Strengths:**

1. the studied problem, jailbreaking LLM, is an important topic of AI safety.
2. although applying DRL to text optimization is not novel, the way how different components are designed and put together in this work is great and interesting.
3. the efficacy improvement of the proposed method is large in some cases but not all.

**Weaknesses:**

1. For black-box evaluation, GPT-3.5-turbo may be dated. GPT-4 is recommeded.
2. I am a little concerned about the results of GPT-Judge since it shows an inconsistent scale compared to Sim. For example, for Llama2-70b-chat in Tab. 1, the performance improvement of the proposed method indicated by GPT-Judge over the previous methods is dramatically larger than that indicated by Sim. The similar cases also occur in Tab. 2 an Tab. 3.
3. The training of RL agent may be instable. How many runs of training did authors test to report the results e.g. in Tab.1. How much is the variance?
4. For the results under Sim., the proposed method sometimes underperform the previous works, e.g., on Mixtral in Tab.1. Similar situations also occur in Tab.2 and Tab.3. Do authors have some insights about why?
5. The efficiency the proposed method is a concern. As shown in Tab. 8, the proposed method, even though more efficient than AutoDAN and GCG, costs much more than other blackbox methods on many models.

**Questions:**

see the Weakness above.

**Limitations:**

limitations are discussed.

---

> ### Author Rebuttal · Authors · 2024-08-07
>
> **The reviewer suggests that GPT-3.5 Turbo may be outdated for black-box evaluation and recommends using GPT-4 instead.**
>
> We thank the reviewer for this suggestion. We evaluate our attack on GPT-4, following the same experiment setup in Section 4. We select the latest GPT-4o-mini (07/18/2024) as the target model and limit the total queries to 10,000. For baselines, we select GPTFUZZER as its performance is better than the other black-box attacks. The results in Table 6 in the submitted pdf clearly demonstrate RLbreaker’s superior ability to bypass strong alignment compared with baseline. The reason why we did not do a large-scale experiment on GPT-4 is in consideration of cost.
>
> **The reviewer also raises concern about the results of GPT-Judge, noting an inconsistent scale compared to Sim.**
>
> We thank the reviewer for pointing this out. We believe this is because the effective value range for Sim. is smaller than GPT-Judge. As we observed during the experiments, Sim. gives at least a 0.6 score even for a rejected answer. However, the GPT-Judge metric spans a full range from [0, 1].
> We chose different text encoders for GPT-Judge and got consistent observation, as we demonstrated in our ablation study in Section 4.
> Furthermore, we use the bge-large-en-v1.5 from Hugging Face to convert text into embeddings for cosine similarity computations. As noted by the model developers, the typical similarity distribution for this model falls within [0.6, 1], which also verifies our previous observation. Across Tables 1, 2, and 3 in Section 4, we observe that the Sim. for RLbreaker and other baselines exceed 0.6.
>
> We also want to note that as discussed in [1], GPT-Judge evaluates responses in a more comprehensive manner, considering contextual relevance, coherence, and the capacity to meaningfully answer the posed questions.  The significant improvements highlighted by GPT-Judge demonstrate that our RL-guided approach produces responses that are more contextually relevant and accurate. It is also important to note that we have not tailored the GPT-Judge prompt specifically for our attack; instead, we employ a prompt directly sourced from existing published work. This approach ensures the generalizability of our method.
>
> [1] COLD-Attack: Jailbreaking LLMs with Stealthiness and Controllability. Guo et al., ICML 2024.
>
> **The reviewer raises concerns about the instability of training the RL agent, asking how many training runs were conducted to report the results in Tab. 1 and what the variance is.**
>
> We thank the reviewer for this question. Our training runs for the RL agent are constrained by a limit of 10,000 queries to the target model. Specifically, we set the number of forward steps in the environment as 16, i.e., we will update our agent after we forward 16 steps in the environment. We will use the selected trajectories to update the agent with our customized PPO algorithm and the training epoch is 4, thus, the total training runs of our agent is 2500 times. We will include all training details in our appendix in the next version.
>
> To understand the variance of our training process, we trained the agent five times using different random seeds on the target model llama2-7b-chat with GPT-3.5-turbo as the helper model. The results in Table 7 and Figure 1 in the submitted pdf show low variance during training and testing across these seeds. We believe the reason is that we constrain the action space. Instead of letting the model choose a token within the vast token space, we let the model mutate the input templates, which gives a much smaller search space and thus more stable agents.
>
> **The reviewer points out that under Sim., our proposed method sometimes underperforms compared to previous works, such as on Mixtral in Table 1, with similar observations made in Tables 2 and 3.**
>
> We thank the reviewer for pointing this out. We suspect that this is because the Mixtral model answers the questions in a way different from the reference answers (given by the Vicuna model). When comparing the answers given by the Mixtral model with reference answers, it is possible that some answers are given a low Sim. because Mixtral answers a question in a very different way from the reference answer. However, when using GPT-Judge, the attack success rate is higher.
>
> As we have discussed in Section 4, Sim. indeed can introduce false negatives when the target model answers a question in a very different way from the reference answer. This is the reason why we used multiple metrics in the evaluation. However, we still use Sim. during the training mainly because it is more efficient than GPT-Judge and introduces fewer false negatives than keyword matching.
>
>
> **The reviewer raises concerns about the efficiency of our proposed method.**
>
> We thank the reviewer for the question. Tab. 8 in our paper shows that RLbreaker is more efficient than AutoDAN and GCG and comparable to GPTFUZZER. Although our method is slower than PAIR and Cipher, these two methods are way less effective than ours. We designed an additional experiment to show the efficiency of our method over those baselines. To make these two methods have comparable performance with our method, we need to significantly increase the total number of queries. Specifically, for Cipher, we executed their jailbreaking prompts up to 50 times per question, considering it as a success if any trial resulted in a successful jailbreak. This led to 16,000 queries at most. For PAIR, we set their two key hyper-parameters: number of iterations and number of streams as 3 and 20 separately, resulting in a maximum of 19,200 queries. Results in Table 8 in the submitted pdf file show that these two methods are actually less efficient than ours if we aim to achieve a similar attack effectiveness.

---

> > ### Comment · Reviewer_5ZZd · 2024-08-11
> >
> > Thanks much for your thorough responses. Most of my concerns have been addressed, so I decide to raise my score to 6. However, I still have concern about the faithfulness of evaluation metric, GPT-Judge. I understand that this is a previously accepted practice. I personally question it because of the existing observation on the instability of LLM in answering the question when it is written in different prompts of the same meaning like [1].
> >
> > [1] Zong et al., Fool Your (Vision and) Language Model With Embarrassingly Simple Permutations, ICML 2024.

---

> > > ### Author Response · Authors · 2024-08-12
> > >
> > > We sincerely appreciate the reviewer for updating the score! We are happy that our response could help address the reviewer's concern.
> > >
> > > Regarding the reliability of the GPT-Judge metrics,  we agree with the reviewer that it is not an entirely stable metric for assessing the success of a jailbreaking attack. However, as shown in both [1] and [2], GPT-4 demonstrates a high correlation with human judgment (0.9), suggesting its potential utility as a reliable verifier for responses provided by victim LLMs. While human annotation remains a more accurate method of judgment, it is costly given our experimental setup, which requires evaluating 320 questions in the test set for 5 baselines and our method. Therefore, we believe GPT-Judge serves as a cost-effective and efficient alternative for evaluating attack effectiveness. Furthermore, we ensure consistency in our evaluations by applying the same metrics across all baseline and our method.
> > >
> > > In our future work, we will carefully consider the reviewer's suggestions and explore more strategies to enhance the reliability and stability of our judgment metrics. For example, integrating results from a harmful content classifier [3] alongside GPT-Judge, or grouping evaluations from multiple LLMs and perform a majority vote.
> > >
> > > [1] COLD-Attack: Jailbreaking LLMs with Stealthiness and Controllability. Guo et al., ICML 2024.
> > >
> > > [2] AutoDAN: Generating Stealthy Jailbreak Prompts on Aligned Large Language Models. Liu et al., ICLR 2024.
> > >
> > > [3] HarmBench: A Standardized Evaluation Framework for Automated Red Teaming and Robust Refusal. Mazeika et al., arXiv 2024.

---

### Official Review · Reviewer_ipyX · 2024-07-13

**Soundness:** 2
**Presentation:** 2
**Contribution:** 2
**Rating:** 5
**Confidence:** 4

**Summary:**

This paper proposes a new jailbreaking attack on LLMs with deep-reinforcement learning (DRL) techniques, which takes jailbreak prompts as states and mutations as actions.

**Strengths:**

1. This paper is the first to leverage DRL techniques to jailbreaking LLMs, bringing new insights to this community.
2. The experiments include multiple LLMs and attack baselines.
3. The evaluation considers ablation studies on each part of the proposed method, showing the robustness against hyper-parameters of the DRL training.

**Weaknesses:**

1. The organization of this paper can be substantially improved to polish readability. For example, Section 2 failed to discuss the background (deep) reinforcement learning and their related work on jailbreaking.
2. Some technique details of the method are not specified. For example, in line 164, which mutators are used are not clearly introduced. Instead, the authors simply use a reference here.
3. The plausibleness of the reference answer $\hat u_i$ derived from an unaligned LLM should be further justified. Specifically, the weakly aligned vicuna-13b does not always respond to harmful prompts. What happens if both the target LLM and this model refuse to answer a harmful query?
4. The experiment does not indicate the training/inference time required for the method.
5. The experiment results show that the improvement of the proposed method over existing methods is somewhat limited.

**Questions:**

See weaknesses.

---

> ### Author Rebuttal · Authors · 2024-08-07
>
> **The reviewer suggests improving the paper's organization to enhance readability. For example, Section 2 lacks discussion on the background of DRL and related work on jailbreaking.**
>
>
> We thank the reviewer for the suggestions. Due to the space limit, we did not add a background section for DRL. Instead, we introduced the key formulations of our DRL system in Section 3. We will add a background of DRL (in the appendix) in the next version. Regarding related works on jailbreaking, we would like to kindly point out that we have included a comprehensive literature review in Section 2, with a special focus on black-box attacks.
>
> To the best of our knowledge, there are no existing works that specifically apply DRL to jailbreaking attacks, although other works apply RL to attack LLMs with different attack goals and threat models, as discussed in Section 2. All these works used RL to directly generate tokens, which is similar to the token-level action baseline in our ablation study (denoted as Token in Figure 2 in our paper). As we demonstrated in our experiments, the action space is too huge for this type of method to work in jailbreaking problems.
>
>
> **The reviewer also suggests that the plausibility of the reference answer from an unaligned LLM needs further justification. The reviewer questions what would occur if both the target LLM and this model refuse to answer a harmful query.**
>
> We thank the reviewer for their questions. First, we would like to clarify that we utilize an uncensored version of the Vicuna-7b model instead of a weekly aligned Vicuna-13b to obtain reference answers. This uncensored Vicuna-7b model has not been fine-tuned with human preferences and lacks guardrails, ensuring it responds to user prompts irrespective of potential harmfulness. We also manually double-check all the reference answers and ensure they are indeed answering the questions.
>
> Additionally, we assume that even if there are any questions that are rejected or deemed irrelevant, the RL agent can still learn an effective strategy about how to select the mutators during the interaction with other questions. To test this assumption, we randomly marked 10% and 20% of the reference answers as unavailable, by setting them as ''I’m sorry I cannot assist with this request''. This also mimics the case when the unaligned model refuses to answer the question. Then we trained our RL agent on Llama2-7b-chat. The results in Table 4 in the submitted pdf demonstrate that even though some of the reference answers are unavailable, our method still achieves good performance on jailbreaking the model.
>
> We will address these points in our next version and we will publish all reference answers used in our experiments upon acceptance.
>
> **The reviewer raises concern that some technical details of the method are missing, e.g., details of mutators, and the training/inference time required for our method.**
>
> We thank the reviewer for the question. We would like to kindly point out that in Section 3, we detail the five mutators we use, including their name and how we conduct the mutation. We also include the LLM that we use to perform the mutation. The prompts for each mutator are shown in Tab. 4 in our paper. Regarding the training and inference time, we consider the total time (training + inference) as one of our efficiency metrics, as we discussed in Section 4, and the results are in Tab. 8 in our paper. We also separately report the training and inference time of our method in Table 5 in the submitted pdf. We will include these results in our next version.
>
>
> **The reviewer comments that the experimental results show our method has limited improvement over existing methods.**
>
> We thank the reviewer for the question. In our experiments, we conducted a comprehensive comparison of our method against five SOTA baselines. These comparisons and metrics were carefully designed to fairly evaluate the effectiveness of our methods.
>
> To address the reviewer’s concern specifically:
>
> Regarding the baselines we choose, we carefully select the four SOTA black-box jailbreaking attacks that cover the genetic method-based attacks and in-context learning-based attacks. We also include the representative white-box attack: GCG. Then for the evaluation metric, we leverage four metrics for a comprehensive attack effectiveness evaluation while existing works usually only leverage one or two. Among those metrics, we consider that a higher GPT-Judge score is enough to show the advantages of our methods over other attacks, as GPT-Judge has a higher correlation with human annotations, as demonstrated by [1]. We also directly adopt their judging prompt to ensure a fair comparison, it also guarantees that we are not designing our attack specifically our own designed prompt.
>
> Our method achieved a significant improvement in the GPT-Judge score, which is a critical metric for assessing the effectiveness of the attacks. Specifically, our approach showed about 40% improvement on Llama2-70b-chat over the best-performing baseline. This is a substantial enhancement, demonstrating the efficacy of our proposed attack. Our attack also achieves the highest GPT-Judge score across all six target LLMs, compared to baseline methods. We also have provided detailed tables (Tab.1 and Tab.6  in our paper) that explicitly illustrate these advancements. We also performed a thorough ablation study to validate the core designs of RLbreaker.
>
> [1] COLD-Attack: Jailbreaking LLMs with Stealthiness and Controllability. Guo et al., ICML 2024.

---

> > ### Author Response · Authors · 2024-08-13
> >
> > Thanks to the Reviewer ipyX again for the insightful comments and questions. Since the discussion phase is about to end, we are writing to kindly ask if the reviewer has any additional comments regarding our response. We are at their disposal for any further questions. In addition, if our new experiments address the reviewer's concern, we would like to kindly ask if the reviewer could reconsider their score.

---

> ### Comment · Reviewer_ipyX · 2024-08-13
>
> Thanks for the rebuttal, your efforts are truly appreciated. I've raised my rating to 5. Some comments:
> - I know that there are no existing works that specifically apply DRL to jailbreaking attacks (see strengths); what I requested here is to discuss how (deep) reinforcement learning has been used to attack ML models / LLMs. Such connections can help the readers better understand to what extent these techniques have been leveraged in the adversarial ML area.
> - The limitation on the requirement of manually double-checking all the reference answers is a limitation that needs to be explicitly acknowledged.
> - It would be great to evaluate stronger attack/defense baselines, like the demonstration-based in-context attack/defense (https://arxiv.org/abs/2310.06387).

---

> > ### Author Response · Authors · 2024-08-13
> >
> > Thanks to the Reviewer ipyX for their insightful comments. We really appreciate the reviewer for raising their score to acknowledge our effort. Below, we would like to provide some clarifications for the reviewer's additional comments.
> >
> > 1.  We totally agree with the reviewer that discussing how DRL has been used in attacking LLMs if not for jailbreaking purposes is super helpful. We have a short summary of existing works in Section 2.  We also experiment with a baseline (token-level RL) that is generalized from existing works in our evaluation. We will further emphasize these in our paper.
> >
> > 2. We would like to clarify that we did a manual check just to assess the quality of the unaligned models in generating reference questions. This is a one-time effort. It is like an ablation study or sanity check and is not required to run our method.
> >
> > 3. Moving forward, we will follow the reviewer's suggestion to add new baselines, including the one pointed out by the reviewer.

---

### Official Review · Reviewer_pYsF · 2024-07-22

**Soundness:** 2
**Presentation:** 2
**Contribution:** 2
**Rating:** 6
**Confidence:** 3

**Summary:**

This paper introduces RLbreaker, a novel deep reinforcement learning (DRL) approach for generating jailbreaking prompts to attack large language models (LLMs). The authors frame jailbreaking as a search problem and design a DRL agent to guide the search process more efficiently than existing stochastic methods. Key technical contributions include a customized reward function, action space based on prompt mutators, and a modified proximal policy optimization (PPO) algorithm. Through extensive experiments, the authors demonstrate that RLbreaker outperforms state-of-the-art jailbreaking attacks across multiple LLMs, shows resilience against existing defenses, and exhibits strong transferability across models. The paper also includes ablation studies validating the key design choices.

**Strengths:**

- Good results on Llama-2-70B (52.5% ASR).
- The proposed method makes sense, and, to the best of my knowledge, no one has proposed using an RL algorithm for guiding jailbreak search.
- The transferability of the attack is non-trivial.

**Weaknesses:**

I have concerns about the reported numbers:
- Why is GCG shown as N/A for GPT-3.5 Turbo? Evaluating the result of a transfer attack would still make sense (and I would expect that the ASR according to GPT-4 as a judge should be above 50%).
- Moreover, why does PAIR perform so badly on GPT-3.5 Turbo? You report 9% ASR while the original paper reports 60% ASR (but on a different set of AdvBench requests). Similarly, Mixtral is known to be a non-robust model, so it’s surprising to see that PAIR achieves only 15% ASR on it.
- Why does the ASR according to the GPT judge is significantly higher when the perplexity filter is enabled (Table 2: 69.1%, Table 1: 52.5%)?

Other concerns:
- Minor: “However, the random nature of genetic algorithms significantly limits the effectiveness of these attacks.” - why?
- Minor: Tables 2 and 3 have a too small font.

I'd be willing to increase the score if the concerns about the reported numbers are resolved.

**Questions:**

See the questions about the reported numbers.

**Limitations:**

Yes.

---

> ### Author Rebuttal · Authors · 2024-08-06
>
> **The reviewer raised concerns regarding the N/A values from GCG for GPT-3.5 Turbo in Tab.1. It could be run on other LLM and obtain the adversarial prompts to do a transfer attack.**
>
> We thank the reviewer for the suggestion. Following the suggestions, we added an experiment to test GCG's performance on GPT-3.5 Turbo. We followed the original GCG setup and used two different models (Vicuna-7b, Vicuna-13b) as the source models to generate the jailbreaking suffixes. Then, we used these suffixes to attack GPT-3.5 Turbo. We followed the same setup in our Section 4 and set the upper bound for the total query times as 10,000. The results in Table 1 in the submitted pdf indicate that when making a transfer attack on GPT-3.5 Turbo, GCG cannot outperform RLbreaker.
>
> **The reviewer then raised concerns regarding why PAIR’s ASR on GPT-3.5 Turbo is lower than what is reported in the original paper and why its ASR on Mixtral is only 15%.**
>
> Thanks for the question. The main reason for a low PAIR performance is that we set an upper bound for the total query times as 10,000 for the entire attack process. The reason is to enable an apple-to-apple comparison for all methods. Given that some methods have a training process but others do not. For PAIR, this limit maps to the action that we set the 10,000 queries for all 320 questions (because PAIR does not have a training process. So all the queries are directly applied to the testing phase). Following this constraint, we set their two key hyper-parameters: number of iterations and number of streams (the number of parallel conversations) as 5 and 6 separately, while they set the number of streams as 20 in the original paper and number of iterations as 5. PAIR also mentioned that the total number of queries had a significant influence on the performance of jailbreaking attacks. Thus this reduction in parallel streams is a primary factor contributing to the lower ASR observed in our results compared to its original paper. Second, we followed [1] and used a different way of writing the GPT-Judge prompt from PAIR.
>
> We added a new experiment that follows the same setup with the PAIR, setting the number of iterations as 5 and the number of streams as 20, which will lead to 32,000 upper bound for the total number of queries. We then ran PAIR and RLbreaker with the new upper bound. We selected Mixtral-8*7B and GPT-3.5-Turbo as the target model and we reported the ASR measured by our GPT-Judge metric and JUDGE function proposed by PAIR. Results in Table 2 in the submitted pdf file demonstrate that RLbreaker still outperforms PAIR across two different evaluation metrics.
>
> [1] COLD-Attack: Jailbreaking LLMs with Stealthiness and Controllability. Guo et al., ICML 2024.
>
> **The reviewer asked why the ASR using GPT judge is higher when the perplexity filter is enabled.**
>
> We thank the reviewer for the question. We would like to kindly point out that for Table 1 in our paper, we are evaluating Llama2-70b-chat, while for Tab. 2, we focus on Llama2-7b-chat. To address the reviewer's question further, in Table 3 in the submitted pdf, we showed the ASR for Llama2-70b-chat model specifically when a perplexity filter is applied. The result aligns with Tab. 2 in Section 4, demonstrating RLbreaker’s resiliency against the perplexity defense.
>
> **The reviewer also asked why the random nature of genetic algorithms significantly limits the effectiveness of these attacks**
>
> We thank the reviewer for raising this insightful question. As we briefly discussed in Section 3, the limitations of genetic algorithms in developing jailbreaking attacks are two-fold.
> Inefficiency in Search Process: Stochastic search methods, including genetic algorithms, initiate with a randomly chosen initial region and explore this region randomly before moving to other areas. This process involves random mutation and selection, which leads to a highly inefficient search process. As demonstrated in the grid search example in Appendix B.2, stochastic search requires at least three times more grid visits compared to guided search, highlighting its inefficiency.
> Constraints of Random Mutation: In the context of jailbreaking attacks, existing methods that employ genetic algorithms iteratively generate new prompts by randomly selecting mutators to modify the current prompts. This randomness in mutator selection significantly constrains the search efficacy, as it often directs computational resources toward less promising areas of the search space. This approach is particularly ineffective in the expansive search spaces common in jailbreaking scenarios. Furthermore, after each selection of the mutators, the absence of informative feedback means that those genetic algorithm-based attacks cannot effectively utilize prior knowledge or feedback. In contrast, DRL-guided searches benefit from RL agents that prioritize actions leading to successful outcomes, driven by the accumulation of rewards.
>
> As a result, the random nature of genetic algorithms limits their effectiveness in jailbreaking attacks primarily due to their inefficient exploration of the search space and the significant computational overhead involved in randomly selecting mutators. This inefficiency is especially problematic in large search spaces, leading to constrained search efficacy and reduced overall effectiveness.
>
> We will add more discussion in the next version.
>
> **Other minor issues.** We thank the reviewer’s suggestion and we will increase the font size of Table 2 and Table 3 in our next version.

---

> > ### Comment · Reviewer_pYsF · 2024-08-12
> > **Follow-up comment**
> >
> > Thanks for the detailed response, it addresses my concerns. I increase my score to 6.

---

> > > ### Author Response · Authors · 2024-08-12
> > >
> > > We would like to thank the reviewer for increasing the score and we are happy that our response could help address the reviewer's concern. As we proceed with the revision, we will be mindful of the suggestions and present a stronger version of our paper based on the reviewer's feedback.

---

### Author Rebuttal · Authors · 2024-08-07

We thank the reviewers for the constructive feedback. Below, we summarize our responses:

**New experiments:**

We added all experiments suggested by reviewers (All the results are in the submitted PDF). Below, we give a brief summary.

1. **GCG transfer attack on black-box model.** We demonstrated the jailbreaking effectiveness of RLbreaker compared to GCG on GPT-3.5-Turbo. We followed GCG and used Llama2-7b-chat to generate jailbreaking suffixes for GCG (Reviewer pYsF).

2. **Compare our method with baselines under a higher query limit (in response to why existing methods have a low performance).** We demonstrated the jailbreaking effectiveness of RLbreaker vs. PAIR on Mixtral-8*7B-Instruct, with a query limit of 32,000. The effectiveness is measured using GPT-Judge and JUDGE metrics from PAIR (Reviewer pYsF).

3. **Our method on Llama2-70b-chat when perplexity defense is added.** We added the results of Llama2-70b-chat when perplexity defense is added (Reviewer pYsF).

4. **Robustness against reference answers.** We demonstrated the robustness of our methods when some of the reference answers are not available (Reviewer ipyX).

5. **Training and inference time of our method.** We reported the training and inference time (in minutes) for RLbreaker across six target LLMs (Reviewer ipyX).

6. **Our method’s performance on jailbreaking the latest GPT-4o-mini.** We demonstrated RLbreaker’s effectiveness in jailbreaking the latest GPT-4o-mini model compared to the best-performing baseline (Reviewer 5ZZd).

7. **Training and testing stability.** We demonstrated RLbreaker’s stability across different random seeds, including training curves of mean rewards and the statistics of GPT-Judge score (Reviewer 5ZZd).

8. **Our method’s performance when the reference answers are obtained using different models (in response to why our method has lower Sim. on Mixtral).** We demonstrated RLbreaker’s jailbreaking effectiveness when we vary the model that is used to obtain the reference answers (Reviewer 5ZZd).

9. **Performance on some out-of-distribution question sets.** We demonstrated RLbreaker’s robustness on two out-of-distribution testing question sets (Reviewer 5U1Z).

Below, we also summarize the key points in our responses:

**Reviewer pYsF**

1. We clarified and demonstrated that by setting a larger upper bound for the total number of queries, the baseline method can meet their ASR in their original paper, and our method can also achieve better results.


2. We clarified why the random nature of genetic method-based attacks significantly limits the effectiveness of jailbreaking attacks.


**Reviewer ipyX**

1. We clarified and pointed out our discussion on the related works of jailbreaking attacks and we will follow the reviewer’s suggestion of adding another section about the background of deep reinforcement learning (DRL).

2. We clarified that we have included all required details of our designed mutators and we also reported the training and inference time separately for our method.

3. We clarified why our method introduces significant improvements over existing baselines.


**Reviewer 5ZZd**

1. We clarified why there is an inconsistency between GPT-Judge and Sim.’s scale, highlighting the reliability of GPT-Judge over Sim. and the significant improvements that RLbreaker introduces.

2. We clarified that the increased expense of RLbreaker is worthwhile as our method generates more powerful jailbreaking prompts thus enhancing LLM safety and alignment through effective adversarial training.


**Reviewer 5U1Z**

1. We clarified how we determine our action space by detailing the selection criteria and the underlying methodology.

---

### Decision · Program_Chairs · 2024-09-25

**Decision:**

Accept (poster)

**Comment:**

**Summary:** This paper introduces RLbreaker, a novel deep reinforcement learning approach to improve the efficiency of jailbreaking attacks on LLMs. The method is evaluated against SOTA baselines across various models, with the results indicating significant improvements in attack success rates. However, the paper has raised concerns regarding the robustness of the evaluation, the stability of the metrics used, and the scalability of the approach.

**Strengths:**
- The idea of applying DRL to guide the jailbreaking process is novel and addresses the inefficiencies of traditional stochastic search methods.
- The empirical results demonstrate that RLbreaker outperforms existing methods on several benchmarks, with clear improvements in attack success rates.
- The paper is well-organized, and the authors provide detailed ablation studies to validate the key design choices of their method.

**Weaknesses:**
- Evaluation Scope: The experiments are conducted on a relatively small test set (300 samples) and on models that are not the latest (e.g., GPT-3.5-turbo). Reviewers expressed concerns about the robustness and generalizability of the results.
- Metric Stability: The reliability of the GPT-Judge metric was questioned, especially given its inconsistency with the Sim. metric in some cases. The authors provided explanations but the concerns remain partially unresolved.
- Efficiency and Scalability: While RLbreaker is more efficient than some baselines, it is still resource-intensive, and its performance improvements may not justify the additional computational cost in all scenarios.

**Unresolved Issues:**
- Robustness and Generalization: The limited scope of the test set and the absence of evaluations on more diverse and updated models (e.g., GPT-4) leave questions about the robustness and generalizability of the method.
- Metric Reliability: The GPT-Judge metric, though widely used, has been flagged for potential instability. The authors have acknowledged this but have not provided a fully satisfactory resolution.
- Manual Effort: The need for manual verification of reference answers, as part of ensuring the quality of the attack, is a limitation that needs to be explicitly recognized.

**Conclusion:** The paper presents a significant advancement in the field of LLM jailbreaking, with innovative use of DRL and strong empirical results. However, the concerns about evaluation robustness, metric stability, and computational efficiency temper the overall impact. Given the strengths and the potential for future improvements, the paper leans towards acceptance, though these issues should be carefully addressed in future work.